# A Flexible Sandwich Structure Carbon Fiber Cloth with Resin Coating Composite Improves Electromagnetic Wave Absorption Performance at Low Frequency

**DOI:** 10.3390/polym14020233

**Published:** 2022-01-07

**Authors:** Yuanjun Liu, Qianqian Lu, Jing Wang, Xiaoming Zhao

**Affiliations:** 1School of Textile Science and Engineering, Tiangong University, Tianjin 300387, China; liuyuanjunsd@163.com (Y.L.); luqianqian1998@163.com (Q.L.); wjwjwjtg@163.com (J.W.); 2Tianjin Key Laboratory of Advanced Textile Composites, Tiangong University, Tianjin 300387, China; 3Tianjin Municipal Key Laboratory of Advanced Fiber and Energy Storage, Tiangong University, Tianjin 300387, China

**Keywords:** resins, carbon fiber cloth, sandwich structure, graphene, absorbing performance, mechanical property

## Abstract

In order to improve the electromagnetic wave absorbing performance of carbon fiber cloth at low frequency and reduce the secondary pollution caused by the shielding mechanism, a flexible sandwich composite was designed by a physical mixing coating process. This was composed of a graphene layer that absorbed waves, a carbon fiber cloth layer that reflected waves, and a graphite layer that absorbed transmitted waves. The influence of the content of graphene was studied by a control variable method on the electromatic and mechanical properties. The structures of defect polarization relaxation and dipole polarization relaxation of graphene, the interfacial polarization and electron polarization of graphite, the conductive network formed in the carbon fiber cloth, and the interfacial polarization of each part, combined together to improve the impedance matching and wave multiple reflections of the material. The study found that the sample with 40% graphene had the most outstanding absorbing performance. The minimum reflection loss value was −18.62 dB, while the frequency was 2.15 GHz and the minimum reflection loss value compared to the sample with no graphene increased 76%. The composites can be mainly applied in the field of flexible electromagnetic protection, such as the preparation of stealth tent, protective covers of electronic boxes, helmet materials for high-speed train drivers, etc.

## 1. Introduction

With the wide application of various types of electronic equipment and with the communication facilities in many aspects of industrial production and daily life, the problem of electromagnetic pollution is of wide public concern [1,2]. Harmful electromagnetic waves can cause information leakage, interfere with the operation of electronic equipment, threaten human health, and shorten the survivability of weapons on the battlefield [3,4,5]. Preparation of electromagnetic protective materials has become a research focus. Compared with the wave absorbing materials, the shielding materials can cause secondary pollution, thus researchers need to prepare materials with a more absorbing mechanism [6,7]. In the past few decades, a lot of research has focused on wave absorbing materials but has been mainly concentrated in the Super High Frequency, the studies of wave absorbing materials below the Super High Frequency have been fewer or not ideal. However, a large number of electronic devices have been used in these lower frequency bands [8], so that the research at the lower frequency bands of wave absorption materials has become very significant.

In recent years, researchers have studied many absorbing materials. Magnetic loss type metal materials belong to the major categories. They are characterized by high density, unstable chemical properties, small specific surface areas, and a weak absorbing performance at low frequency [9]. Magnetic metal materials often need to be compounded with other materials to improve the absorbing performance. For example, Liu et al. [10] prepared TiO_2_/Ti_3_C_2_Tx/Fe_3_O_4_ composites in different proportions by a simple hydrothermal reaction route. By adjusting the ratio, the two-dimensional materials showed a good microwave absorbing performance in terms of maximum RL value and absorber thickness. Carbon-based absorbing materials have the advantages of variety, thin thickness, low cost, light weight, good corrosion resistance, and wide use, while having a certain absorbing performance in the low frequency band [9,11]. Carbon fiber has the characteristics of low density, good flexibility, high hardness, high temperature resistance, corrosion resistance, good electrical conductivity, etc. [12]. For example, Jin et al. [13] proposed a multi-plate composite radar absorption structure, in which short carbon fiber layers with controllable dielectric constants are evenly and randomly dispersed and inserted between glass fabrics. By measuring the complex dielectric constant, the area density of the short carbon fiber layer is controlled, which can absorb electromagnetic waves in the target frequency band. Carbon fiber cloth made by a textile process is widely used in the aerospace, industry, construction, medical and other fields with the requirement of wave absorption [14]. Due to the overlap between carbon fibers in the carbon fiber cloth, the formed three dimensional continuous conductive network gives carbon fiber cloth an excellent shielding performance, and a weak absorbing property [15]. Preparation of the absorbing layer can effectively improve the absorbing performance of the carbon fiber cloth. Graphene and graphite are widely used as two kinds of absorbing functional particles, with the electromagnetic waves then being attenuated by dielectric loss [16]. Graphene has the advantages of a high specific surface area, thin thickness, light weight, and stable chemical properties [17]. For example, Liu et al. [18] prepared a series of cobalt-doped ferrite/graphene nanocomposites with different graphene contents by a simple one-pot method. The cobalt-doped ferrite particles are uniformly loaded on the surface of graphene nanosheets, which gives the composite good absorbing properties. By adjusting the content of graphene, the dielectric properties and magnetic properties of the nanocomposites can be improved and optimized at the same time, further enhancing the impedance matching and attenuation ability. Graphite was one of the earliest applied wave-absorbing materials, and has many advantages, such as the wide source of raw materials, light weight, and low price.

To improve the absorbing performance of carbon fiber cloth, this paper designed a kind of flexible sandwich structure carbon fiber composite by a coating process with a physical mixing method, composed of a graphene layer, a carbon fiber cloth layer, and a graphite layer. The graphene layer improves the impedance matching and absorbs the wave, the carbon fiber cloth as the support structure reflects the electromagnetic wave into the graphene layer to be re-absorbed, and the graphite layer can re-absorb the electromagnetic wave that is transmitted through the carbon fiber layer, as the transmitted wave is less; the use of graphite reduces the cost significantly. Each layer is bonded by a scraper coating process which has the advantages of a simple process, a controllable process, low cost, and mass production, etc. The adhesive used was PU2540 type polyurethane which is flexible, environmentally friendly, non-toxic, and cheap [19,20]. In the previous study, we found that the composites prepared with 30% (weight percentage) graphite in polyurethane on the surface of carbon fiber cloth had the best absorbing effect. In this paper, the control variable method was used to study the influence of the content of graphene particles in the graphene layer, which play a major role in absorbing the wave, on the shielding performance, and the absorbing performance at a frequency of 0.02–3.00 GHz, the dielectric properties at a frequency of 0.02–1.00 GHz, the electrical conductivity, and mechanical properties of the composites.

## 2. Materials and Methods

### 2.1. Materials

The main experimental material was plain carbon fiber cloth, provided by Weiduowei Technology Co., Ltd., Tianjin, China. Other chemicals included graphite powders, graphene, polyurethane, thickener, and defoaming agent. The graphite powders (≥98.0%, Q/HG3991-88) were purchased from Tianjin Fengchuan Chemical Reagent Technology Co., Ltd, Tianjin, China. Graphene (of fineness 5–15 μm, and purity larger than 95%) from Tianjin Kairuisi Fine Chemical Co., Ltd, Tianjin, China. Polyurethane (PU-2540) from Guangzhou Yuheng Environmental Protection Materials Co., Ltd, Guangzhou, China. Thickener (7011) from Guangzhou Dian Wood Composite Material Business Department, Guangzhou, China. And defoaming agent (1502) from Wuxi Redwood New Material Technology Co., Ltd., Wuxi, China.

### 2.2. Preparation of Materials

Preparation of the carbon fiber cloth: The plain carbon fiber cloth was cut into samples of 50 × 25 cm and fixed on the needle plate of a blade coating machine (produced by Werner Mathis, a Swiss company, LTE-S87609 type), requiring that the base cloth be in a state of tension and with a smooth surface without wrinkles; a uniform tension should be applied on the carbon fiber each time.

Preparation of the coatings: The schematic preparation of the coating is included in Figure 1. First, the polyurethane and the functional particle materials (graphene or graphite) were weighed; next the weighed polyurethane was placed in an agitator, and one of the functional particle materials was added to the polyurethane at a low speed of 600 RPM, after all functional material particles had been added to the polyurethane, the speed of the agitator was uniformly raised to 2000 RPM, and the solution was stirred for 5 min. Next thickener (1–2% of the total weight) and defoaming agent (1–3% of the total weight) were added and the solution was stirred for 35 min, and a well-mixed coating was obtained; The viscosity of the dope was measured with the No. 4 rotor of the Digital Viscometer (produced by Shanghai Hengping Instrument Factory, SNB-2 type) with a rotating speed of 6 RPM and a viscosity range between 30,000 and 40,000 mPa·s.

Preparation of the graphene and graphite layers: The preparation of the sandwich structure can be seen in Figure 1. First, the prepared carbon fiber cloth was placed on the blade coating machine, and its scraper was fixed, then the thickness (thickness = graphite layer thickness + thickness of the base cloth) was adjusted; Next the speed and coating distance of the blade coating machine were adjusted, and an appropriate coating was made on the surface of the carbon fiber cloth. The scraper was removed after the coating process and then the obtained coated fabric was placed in an oven and dried at 80 °C under vacuum for 10 min. After drying, the preparation of the graphite layer was finished, the thickness of the coated material obtained was measured, and the material was reversed. Finally, the scraper was re-fixed and the thickness (thickness = thickness of the graphite layer + the thickness of graphene layer) was adjusted, and the graphene layer was prepared in the same way. Each layer area needed to be no less than 30 × 22 cm.

### 2.3. Test Indicators and Methods

Test for the viscosity: An SNB-2 digital viscometer (made by Shanghai Hengping Instrument Factory, Shanghai, China) was used to measure the viscosity of prepared coatings, and the appropriate rotors and rotation speed were selected according to the range table.

Test for the thickness: A YG141D digital fabric thickness meter (made by Laizhou Electronic Instrument Co., Ltd., Laizhou, China) was used to measure the coating thickness. Multiple measurements were made at different locations on the coating materials, the measured data was recorded, and the mean thickness was calculated to reduce the measurement error.

Test for the shielding effectiveness: A ZNB40 vector network analyzer (made by Rohde & Schwarz, Munich, Germany) was used to measure the shielding effectiveness of the samples. According to the standard of GJB 6190-2008—“measuring methods for shielding effectiveness of electromagnetic shielding materials”—the test frequency range was 0.01–3.00 GHz and the sizes of the samples were 13 cm in diameter [19,20,21,22].

Test for the reflection loss: A ZNB40 vector network analyzer (made by Rohde & Schwarz, Munich, Germany) was used to measure the reflection loss of samples. The test frequency range was 0.02–3.00 GHz, the sample size was a circle with an outer diameter of 7.6 cm and an inner diameter of 3.35 cm [19,20,21,22].

Test for the dielectric properties: The dielectric properties of materials were measured with a BDS50 dielectric spectrometer (made by Novocontrol Gmbh, Frankfurt, Germany) according to the standard of SJ20512-1995—“Test methods for permitivity and permeability of microwave high loss solid materials”.The size of the sample was 2 × 2 cm and the test range was 0.02–1.00 GHz [19,20,21,22].

Test for the surface resistance: The ohmic range of a F8808A desktop digital multimeter (made by Fluke Testing Instrument Co., Ltd., Everett, WA, USA) was used to measure the surface resistance of each sample. The surface resistance of samples per unit length (1 cm) on the samples’ surface was measured, and 20 different locations were continuously selected to carry out the test after the maximum and minimum data had been removed. The average value was taken to reduce the error [19,20,21,22].

Test for the tensile strength: A 3369 INSTRON universal strength machine (made by the American INSTRON Company, Boston, MA, USA) was used to measure the tensile strength of the samples according to the testing method for the tensile properties of the GB1447283 standard, and the size of the samples was 15 × 5 cm.

## 3. Results and Discussion

To improve the wave absorbing performance of carbon fiber cloth at low frequency, a flexible sandwich structure of carbon fiber cloth composite was designed with a physical mixing coating method; the structure is shown in Figure 2d. The composite was composed of a graphene layer absorbing the wave on the surface, a carbon fiber cloth layer reflecting the wave in the middle and a graphite layer re-absorbing the transmitting wave at the bottom. To meet the requirement of a thin absorbing material, the thickness of the graphene and graphite layer was set at 1 mm in the experiment. In the previous experiment, we found that only when the graphite absorbing layer was prepared on the surface of the carbon fiber cloth, had the composite with 30% graphite in polyurethane a better electromagnetic absorbing effect, while the content of the graphene in the layer had a great influence on the absorbing performance of the composite. Therefore, five kinds of composites with different graphene contents were prepared by the control variable method for the experiment. The specific technological parameters are shown in Table 1. First, the influence of the content of graphene on shielding and absorbing performance was investigated at a frequency of 0.02–3.00 GHz, and the conductivity performance was observed. The experiments showed that graphene content has a little influence on the shielding properties between the frequency of 0–1 GHz, and the wave absorbing performance was enhanced significantly. To study the absorbing mechanism of this frequency, the dielectric properties of composites in the frequency range of 0.02–1.00 GHz were studied. With the excellent mechanical properties of the carbon fiber cloth, the effects of the content of graphene on the mechanical properties were investigated.

### 3.1. The Influence of the Content of Graphene on the Shielding, Absorbing, and Conductive Properties of the Composites

There are two main parameters of electromagnetic properties of electromagnetic protection materials, namely shielding efficiency (SE) and reflection loss (RL) value. The former represents the shielding performance of the composite to electromagnetic waves and its value is positive; the larger the value, the better the shielding performance will be. The latter represents the absorbing performance to electromagnetic waves and its value is negative; the smaller the value, the better the absorbing performance will be. Both are very important for the sandwich structure carbon fiber cloth composite that was designed to improve the absorbing performance in this paper. The conductive property has a certain auxiliary role in the study of the electromagnetic properties of the material.

#### 3.1.1. The Shielding Performance

As can be seen from Figure 2a, within a frequency range of 0.05–3.00 GHz, the values of the shielding effectiveness of samples 1, 2, 3, 4, and 5 showed first decreasing, next increasing, and then a decreasing trend. It may be the result of carbon fibers overlapping with each other to form a conductive network for the flow of carriers, the flowing carriers then interacting with the electromagnetic field to shield electromagnetic waves [23]. The absorbing performances of graphite and graphene were limited at this frequency wave band and the shielding performance of the materials was excellent. As the frequency increased, the shielding ability of the material to electromagnetic waves was gradually enhanced, the frequency continued to increase, the amount of incident electromagnetic waves gradually increased, and the amount of electromagnetic waves that could be shielded was gradually saturated until the maximum was reached. However, with further increasing frequencies, the electromagnetic waves that could not be shielded transmitted the composite, and thus the shielding ability showed a gradually weakening trend. The growth of the content of graphene led to an increase of the amounts of electrons, ions, and inherent dipoles, and the probability of graphene particles contacting with each other became larger, the conductive network inside the material was denser, and the conductivity was better. Thus, with the increase of electromagnetic wave frequency, samples with more graphene content tended to have a peak earlier and a higher peak. The shielding efficiency peak of sample 5 with the highest graphene content was 69.89 dB when the wave frequency was 1.53 GHz. In a word, compared with control sample 1, samples 2, 3, 4, and 5 showed improved shielding efficiency in the narrow band range, but it was decreased in others.

#### 3.1.2. Absorbing Performance

As can be seen from Figure 2b, the reflection loss values of all samples fluctuated with the increase of electromagnetic wave frequency in the range of 0.02–1.25 GHz. Compared with sample 1, the absorbing ability of the other samples in this frequency range improved, corresponding with the shielding efficiency diagram in Figure 2a, The shielding ability was being reduced while the absorbing performance was being improved, and the electromagnetic wave was transformed into other energy, mainly heat energy. In the range of 1.25–3.00 GHz, the absorbing performance of samples 1, 2, 3, and 4 tended to be stable, the absorbing performance of samples 3 and 4 were slightly improved compared with sample 1, and the minimum reflection loss values of sample 5 were greatly improved. Sample 5 had the best absorbing performance, and the value was improved of about 76% compared to that of sample 1. The reflection loss value of less than −5 dB almost took up 1/3 of the whole test range which was 0.75 GHz more than sample 1. Graphene has a unique wave absorbing property due to the phenomena of electronic dipole polarization–relaxation and structural defective polarization-relaxation [24,25]. Graphite is one type of electrical loss absorbing agent with a large dielectric loss tangent value, which can absorb electromagnetic waves according to interface polarization attenuation or electronic polarization of the mediums [26,27]. Carbon fiber cloth has almost no wave-absorbing property, and thus the ability to absorb electromagnetic waves is weaker; while the two absorbing particles compound with carbon fibers respectively to form the layer interface. The weak wave absorbing ability of carbon fiber cloth can be enhanced due to excellent impedance matching between the carbon fibers and graphite or graphene. Among them, graphite, graphene and polyurethane, as well as between them and the carbon fiber, all have heterogeneous interfaces. Under the action of an electric field, charge accumulates at the interface of the two heterogeneous materials and the resulting interface polarization loss has a significant attenuation effect on the electromagnetic wave energy [28]. With the increasing content of graphene, the minimum value of the reflection loss became smaller, and the frequency range corresponding to an excellent wave absorbing property became wider. It may be that the increase of the content of graphene led to an increase in the amount of graphene particles in the layer per unit volume. The amounts of electrons, ions, and inherent dipoles were also growing, the impedance matching between carbon fibers and graphene became enhanced, and the ability to absorb electromagnetic waves was improved accordingly.

#### 3.1.3. Conductive Performance

To verify the change of the conductive property of the composites, we measured the surface resistance on a unit length (1 cm) of the graphene layer surface and the test results are shown in Figure 2c. The values of the surface resistance of sample 1 and 2 were extremely large, which exceeded the measuring range of the testing instrument. It may be that the coating of sample 1 was a layer of polyurethane, which is one type of polymer with a stable structure that cannot carry out an electronic transmission. For sample 2, because the content of graphene relative to that of polyurethane was lower, the polyurethane negated the excellent conductivity of graphene. Then, with the increasing contents of graphene, the value of the surface resistance decreased gradually, and the conductivity of the composite also was enhanced gradually. The resistance was still very large, which helped to enhance the absorbing property of the composite.

### 3.2. The Influence of the Content of Graphene on the Dielectric Properties of the Composites

The dielectric properties test is very important for carbon-based wave absorbing materials, as it is an indirect indicator which shows the electromagnetic properties of materials. The dielectric properties test mainly includes the real part of the dielectric constant, the imaginary part of the dielectric constant, and the loss tangent value. The real part of the dielectric constant represents the polarization ability of the electromagnetic wave, the imaginary part represents the loss ability, and the loss tangent value represents the attenuation ability. In this paper, the real part and the imaginary part of the dielectric constant, and the loss tangent value of the sample were tested in the frequency range of 0.02–1.00 GHz, as shown in Figure 3a–c, Figure 3d is an enlarged figure of Figure 3c,e showing the action mechanism of each part of the composite to electromagnetic waves.

It can be seen from Figure 3 that the dielectric properties of sample 1 were not affected by the electromagnetic wave frequency while the other samples were changed. The content of graphene was the main reason for the changes in the polarization, loss, and attenuation capacity of the materials to electromagnetic waves and this may be related to the interfacial polarization between graphene, polyurethane, and carbon fiber. Due to the lower content of graphene in sample 2, each part of the dielectric constant had a small amount of improvement compared with that in sample 1. The values of the other samples with a relatively high content of graphene varied greatly with the change in the electromagnetic wave frequency.

#### 3.2.1. The Real Part of the Dielectric Constant

It can be seen from Figure 3a that compared with sample 1, the real part of the dielectric constant of the other samples increased rapidly at first and then slowly decreased to a steady trend with the increase of electromagnetic wave frequency. This may be the result of an interaction of graphite, graphene, and carbon fibers. The impedance matching of electrons, ions, and inherent dipoles in the composite and the impedance matching of their interfaces were both good, which led to an enhancement of the ability to store charges [29,30,31]. However, with the increasing frequencies of the incident electric field, the effects of the internal structure and impedance matching both reached the upper limit, and the polarization ability to electromagnetic waves reached the upper limit accordingly. As the frequency further increased, the amount of incident electromagnetic waves gradually increased, but the amount of electromagnetic waves that could be polarized was limited, and thus the ability for storing charges weakened gradually. The value of the real part of the dielectric constant of the samples with higher graphene content was larger when the real part of the sample tended to be stable. It may be that the probability of graphene particles contacting with each other became larger with the increasing content of graphene, the gap between particles was smaller, the conductive network inside the material was denser, and the conductivity was better. The impedance matching between carbon fibers and graphene was enhanced, and the polarization ability to electromagnetic waves was also enhanced.

#### 3.2.2. The Imaginary Part of the Dielectric Constant

It can be seen from Figure 3b that compared with sample 1, the value of the imaginary part of the dielectric constant of sample 2 had a trend of first increasing and then slowly decreasing to a stable state, while the value of the other samples fluctuated greatly at a relatively stable state in the measured frequency range. This may be the result of the enhancement of the electronic polarization–attenuation ability of the graphite layer and the electronic dipole polarization–relaxation ability of the graphene layer, and thus the loss of ability to electromagnetic waves was enhanced gradually [32,33]. However, with the increasing frequencies of the incident electric field, the eddy current loss caused by an increase of current gradually dominated, and the positive and negative charges heading off from the original equilibrium position in the layer had to return to the original equilibrium position, but could not keep up with the changing frequencies, and thus the loss ability to electromagnetic waves showed a gradually weakening trend [34,35]. The higher the content of graphene of the samples, the stronger was the loss ability of the electromagnetic wave in this test frequency range. That is because with the increase of graphene content, the eddy current loss inside the material was stronger, and the loss capacity to electromagnetic waves was stronger. Moreover, the impedance match of the sample with 20% graphene content was stronger than that of the sample with 30% graphene.

#### 3.2.3. The Loss Tangent Value

It can be seen from Figure 3c,d that compared with sample 1, the loss tangent value of sample 2 increased somewhat while the loss tangent values of sample 3, 4, and 5 showed a trend of rapid increase at first and then rapid decrease to a steady state. It may be that carbon fibers can be seen as one type of semiconductor material with an excellent conductivity. Internal fibers can overlap with each other to form a conductive network, a component perpendicular to the incident electric field of the carbon fibers and the structure of each absorbing layer can be produced to attenuate electromagnetic waves. As the frequency of the applied electric field increased, the amount of attenuated electromagnetic waves increased gradually [36]. Due to the conductivity of the carbon fibers, the electronic polarization–attenuation ability of the graphite, and the electronic dipole polarization–relaxation ability of graphene, the attenuation ability of the composite to electromagnetic waves within a specific frequency range was greatly enhanced, and attenuated the majority of incident electromagnetic waves inside the material. As the incident frequency increased, the attenuation ability to electromagnetic waves gradually weakened until there remained a stable state. The higher the content of graphene, the greater was the tangent loss value. Sample 5 with the highest graphene content has the best attenuation ability for the electromagnetic wave, and its loss tangent value reaches 304.85. With the increase of graphene content, the number of electrons, ions, and intrinsic dipoles, and the attenuation ability of the electromagnetic waves became enhanced [37,38]. The attenuation ability of sample 3 to electromagnetic waves was better than that of sample 4, which was similar to Figure 3b. It is possible that the impedance matching characteristic of sample 3 was better than that of sample 4.

### 3.3. The Influence of the Content of Graphene on the Graphene Layer of the Composite on Mechanical Properties

The mechanical properties of the testing samples are shown in Table 2, the strength test machine is shown in Figure 4a and the displacement–load curve is shown in Figure 4b. As can be seen from Table 2 and Figure 4b, the content of graphene had little effect on the tensile strength, which indicates that the prepared composites can improve their absorbing properties while having no deterioration in their tensile properties. Samples 1 to 5 basically met the trend in that the maximum load increased with increasing contents of graphene. The content of graphene became the main factor affecting the tensile property of the composite, because graphene has a single-layer carbon atom structure with all atoms in the same plane, thus it has good toughness, excellent strength, and a unique deformation mechanism. This unique deformation mechanism can cause the hexagonal structure of graphene in the layer to be destroyed during the stretching process, and eventually lead to a tensile fracture of the coated material [39]. However, as the content of graphene particles gradually increased, the distribution of graphene particles became more uniform in the composite. Moreover, the higher the content of graphene, the stronger the elastic force and anti-pressure ability of the composite, and thus the maximum load increased with the increasing contents of graphene. The excellent mechanical properties of the material benefit from the joint action of carbon fiber, graphite, and graphene, as well as the special plain woven structure of the carbon fiber.

## 4. Conclusions

In this paper, sandwich structure carbon fiber cloth composites prepared by a coating technology can effectively improve wave absorbing performance at low frequency. The electromagnetic parameters of the composites varied greatly in the different test ranges. The sample with a content of 40% graphene in the polyurethane had the most outstanding absorbing performance; the reflection loss value was −18.62 dB when the electromagnetic wave frequency was 2.15 GHz. The absorbing performance is mainly due to its excellent attenuation and loss ability to electromagnetic waves, and conductivity performance. The design of the sandwich structure did not deteriorate the tensile properties of the composites. The design of the absorbing material has the advantages of a simple process, environmental protection, and low price, while being suitable for industrial production. At low frequency, it has the advantages of thin thickness, light weight, good strength, and flexibility.

## Figures and Tables

**Figure 1 polymers-14-00233-f001:**
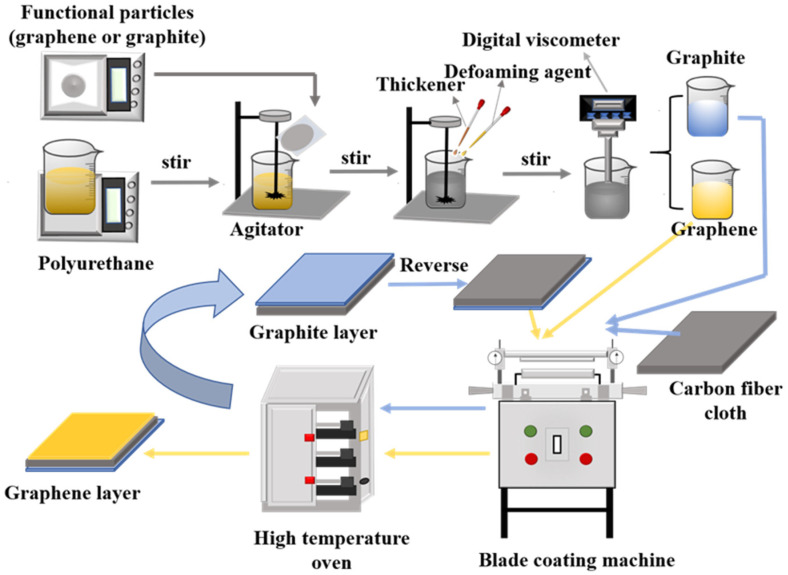
The simple methods for preparing the flexible sandwich structure of carbon fiber cloth.

**Figure 2 polymers-14-00233-f002:**
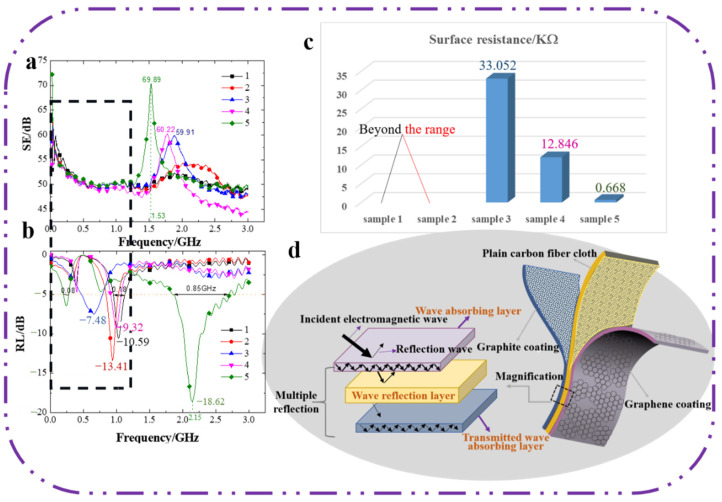
(**a**) The shielding properties of the composites. (**b**) The absorbing properties of the composites. (**c**) The conductive properties of the composites, (**d**) The structure model.

**Figure 3 polymers-14-00233-f003:**
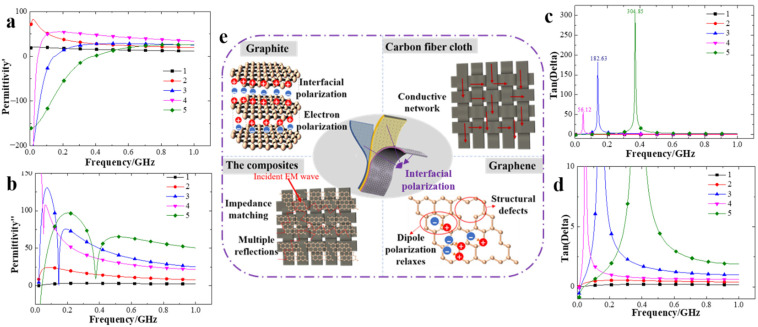
The influence of the content of graphene on dielectric properties of composites. (**a**) The real part of the dielectric constant. (**b**) The imaginary part of the dielectric constant. (**c**) The loss tangent value. (**d**) An enlargement of the loss tangent value. (**e**) The action mechanism of each part of the composite to electromagnetic waves.

**Figure 4 polymers-14-00233-f004:**
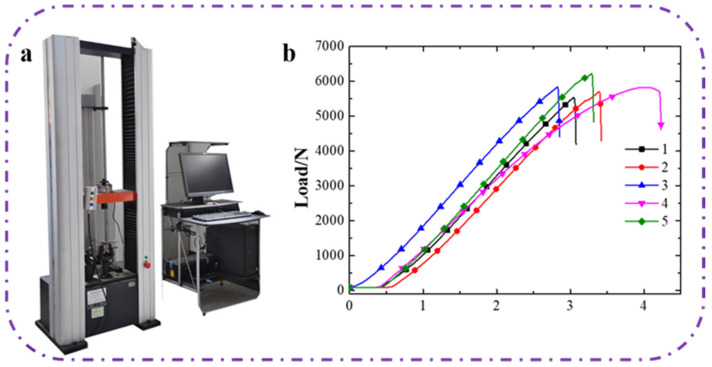
(**a**) A 3369 INSTRON universal strength machine. (**b**) The influence of the content of graphene on the mechanical properties.

**Table 1 polymers-14-00233-t001:** Table for technological parameters.

Sample	Content of Graphene on Wave Absorbing Layer (%)	Content of Graphite on Re-Absorbing Transmitted Wave Layer (%)	Thickness of Each Absorbing Layer (mm)
1	0	30	1.0
2	10	30	1.0
3	20	30	1.0
4	30	30	1.0
5	40	30	1.0

Note: the content of functional particles refers to a percentage of the weight content of functional particles relative to that of polyurethane; the viscosity of each layer of coating was 37,000 mPa·s.

**Table 2 polymers-14-00233-t002:** Parameters of mechanical properties of composites with different contents of graphene.

Sample	Maximum Load (kN)	Maximum Load Displacement (mm)	Maximum Load Tensile Stress (kMPa)	Modulus of Elasticity (kMPa)
1	5.54	3.04	3.79	146.04
2	5.71	3.39	3.56	141.51
3	5.84	2.83	3.64	147.54
4	5.83	3.98	3.63	126.99
5	6.22	3.29	3.62	149.73

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
