# Peer review of "A Flexible Sandwich Structure Carbon Fiber Cloth with Resin Coating Composite Improves Electromagnetic Wave Absorption Performance at Low Frequency"

_polymers, 2022, doi:10.3390/polym14020233_

Round 1

Reviewer 1 Report

Dear Authors,

good paper, congratulation

I have one remark:

Please change the number of digits in table 2. I thinks 3 numbers are enough

5536.43262 to 5.54 kN

BR

The reviewer

Author Response

Dear Prof. Reviewer,

    On behalf of my co-author, we thank you very much for giving us an opportunity to revise our manuscript(polymers-1494143), we appreciate you very much for your positive and constructive comments and suggestion on our manuscript entitled “The Flexible Sandwich Structure Carbon Fiber Cloth with Resins Coating Composite Improves the Electromagnetic Wave Absorption Performance at Low Frequency”.

     I would like to express my sincere thanks for your affirmation for our manuscript and constructive comments. I had reviewed them carefully and the revisions have been made. All the changes in the revised submission were marked in red shadow for easy tracking. The responses for your comments are as follows

Comment 1: Please change the number of digits in table 2. I thinks 3 numbers are enough 5536.43262 to 5.54 kN.

Response: Thank you for your advises. we have changed the number of digits in table 2 as follows. This suggestion adds to the simplicity of the article. Thank you again sincerely.

Table 2. Parameters of mechanical properties of composites with different contents of graphene.

Sample

Maximum load (kN)

Maximum load displacement (mm)

Maximum load tensile stress (kMPa)

Modulus of elasticity (kMPa)

1

5.54

3.04

3.79

146.04

2

5.71

3.39

3.56

141.51

3

5.84

2.83

3.64

147.54

4

5.83

3.98

3.63

126.99

5

6.22

3.29

3.62

149.73

     We appreciated your consideration of our manuscript, and we look forward to receiving your comments. Please acknowledge receipt of this manuscript at your convenience, and let us know if you need any further information.

Sincerely yours,

Xiaoming Zhao

Reviewer 2 Report

Authors reported the synthesis of flexible sandwich structure of carbon fiber cloth by using combination methods, and investigated their structural and electromagnetic absorbing properties. This work has certain reference function as an applied research. Some issues should be addressed before acceptance.

1, Authors should figure out the significance and real-life application of the present paper in Abstract section.

2, In Figure 2, the Figure 2d is missing.

3, Is the material (cloth composite) obtained from physical mixing or in-situ growth? It’s not mentioned in the manuscript. There is no way to judge physical mixing or in-situ growth by the current data. The effect of the interfacial polarization to improve the microwave absorption will be neglected if the material just be obtained by physical mixing.

4, Some key and important research results about 2D absorbers should be mentioned and cited so that we can provide a solid background and progress to the readers, such as Journal of Materials Chemistry C, 2016, 4, 9738; Composites Part A, 2018, 115, 371; Giant, 2021, 8, 100076.

5, In electromagnetic wave absorption section, interfacial polarization often happens at heterogeneous interfaces rather than homogeneous medium. Please investigate the mechanisms deeply.

6, Herein, the preparation method of absorbers is hot pressing? The percentage of absorber filler should be introduced. In addition, as far as I am concerned, it is just to meet the test conditions of coaxial line, and how to achieve the strength requirements in practical application, please give more details.

Author Response

Dear Prof. Reviewer,

    On behalf of my co-author, we thank you very much for giving us an opportunity to revise our manuscript(polymers-1494143), we appreciate you very much for your positive and constructive comments and suggestion on our manuscript entitled “The Flexible Sandwich Structure Carbon Fiber Cloth with Resins Coating Composite Improves the Electromagnetic Wave Absorption Performance at Low Frequency”.

       I would like to express my sincere thanks for your constructive comments. I had reviewed them carefully and the revisions have been made. All the changes in the revised submission were marked in red shadow for easy tracking. The responses for your comments are as follows

Comment 1: Authors should figure out the significance and real-life application of the present paper in Abstract section.

Response: Thank you for your advice. The significance of this paper is mainly to improve the absorbing performance of carbon fiber fabric, reduce the secondary pollution caused by shielding mechanism, which can be mainly applied to the field of flexible electromagnetic protection, such as the preparation of stealth tent, protective cover of electronic box, helmet materials for high-speed train drivers and so on. This suggestion can help readers better understand the realistic meaning of the article, and I have included it in the abstract section. According to the requirements of the journal, the abstract does not exceed 200 words. Therefore, after adding content, some deletions have been made to other parts. Thank you very much again.

Comment 2: In Figure 2, the Figure 2d is missing.

Response: Thank you for pointing out the mistake. We are very sorry that the mistake that should have been "(d)" was written into "(e)" in Figure 2 due to our carelessness, which has been corrected in the original text. It is necessary to correct every mistake in the paper. We have also repeatedly checked the full text to try not to make such mistakes again. Thank you again sincerely.

Comment 3: Is the material (cloth composite) obtained from physical mixing or in-situ growth? It’s not mentioned in the manuscript. There is no way to judge physical mixing or in-situ growth by the current data. The effect of the interfacial polarization to improve the microwave absorption will be neglected if the material just be obtained by physical mixing.

Response: Thank you for your quenstion and advice. The material is made by physical mixing method, the preparation of composite materials was shown in Figure 1 of this paper.Firstly, putting the functional particals of graphite or graphene into the polyurethane at a high stirring speed. Then, the mixture of functional particles and polyurethane was coated on both sides of the carbon fiber fabric through controlling the thickness of the coating machine. Finally, the coated carbon fiber composite can be prepared by drying the fabric. The experimental method has been specially explained in the abstract, introduction and experiment part of the original text.

     Among graphite, graphene and polyurethane, as well as between them and carbon fiber, all belong to heterogeneous interface, which can attenuate and absorb electromagnetic waves because of the interface polarization. This is very important for the study of this material, which has been supplemented in the abstract and results and discussion section of 3.1.2 and 3.2 part, and change the Figure 3e. Thank you again sincerely.

Comment 4: Some key and important research results about 2D absorbers should be mentioned and cited so that we can provide a solid background and progress to the readers, such as Journal of Materials Chemistry C, 2016, 4, 9738; Composites Part A, 2018, 115, 371; Giant, 2021, 8, 100076.

Response: Thank you for your advice. We have added the background description in the second paragraph of the introduction part, and quoted the first two articals you listed and some other relevant articals. I am really sorry that I could not find the third artical you listed, which may be related to the database. It is necessary to introduce the background of previous studies. Thank you again for your suggestion.

Comment 5: In electromagnetic wave absorption section, interfacial polarization often happens at heterogeneous interfaces rather than homogeneous medium. Please investigate the mechanisms deeply.

Response: Thank you for your suggestion. I think this question is similar to the second half of comment 3, so I answered them together and added them in the original text which has been supplemented in the abstract and results and discussion section of 3.1.2 part that is “And among them, graphite, graphene and polyurethane, as well as between them and carbon fiber, all belong to heterogeneous interfaces. Under the action of electric field, charge accumulates at the interface of the two heterogeneous materials, and the resulting interface polarization loss has a significant attenuation effect on electromagnetic wave energy.” and 3.2 part that is “this may be related to the interfacial polarization between graphene, polyurethane and carbon fiber.”, and change the Figure 3e. Interface polarization is very important in electromagnetic wave absorption at heterogeneous interfaces. Thank you again for your suggestion.

Comment 6: Herein, the preparation method of absorbers is hot pressing? The percentage of absorber filler should be introduced.

Response: Thank you for your question and suggestion. The preparation method is not hot pressing method in this paper, but coating method. The functional particles are added into the flexible PU2540 polyurethane and stirred evenly at a high speed, and then poured into the coating machine that can control the coating speed and thickness, coated on the face of the plain carbon fiber fabric, and dried to get the composite material. The percentage of absorber filler was introduced in the Table 1 of 5 page, this percentage is the weight ratio of functional particles and polyurethane. Thank you again for your question. 

Comment 7: In addition, as far as I am concerned, it is just to meet the test conditions of coaxial line, and how to achieve the strength requirements in practical application, please give more details.

Response: Thank you for your question and suggestion. It's not only satisfy the requirement of the coaxial cable of the test. The carbon fiber cloth is manufactured by textile woven process, that is flexible and can be used directly. Woven process can make the fabric a certain strength, adding with the ultra-high strength of carbon fiber bundle itself, graphene, graphite and polyurethane. The composite has excellent mechanical properties, and the strength test is based on GB1447283 standard in this paper. We added this issue in section 3.3 on page 9. How to meet the strength requirements in practical application is a good question, which will be further developed in the follow-up study. Thank you for your advice.

       We appreciated your consideration of our manuscript, and we look forward to receiving your comments. Please acknowledge receipt of this manuscript at your convenience, and let us know if you need any further information.

Sincerely yours,

Xiaoming Zhao

Round 2

Reviewer 2 Report

All issues were well addressed, and this work has a certain reference function as an applied research.